# A Perturbed Asymmetrical Y-TypeSheathless Chip for Particle Control Based on Adjustable Tilted-Angle Traveling Surface Acoustic Waves (ataTSAWs)

**DOI:** 10.3390/bios12080611

**Published:** 2022-08-07

**Authors:** Junping Duan, Miaomiao Ji, Binzhen Zhang

**Affiliations:** Key Laboratory of Instrumentation Science & Dynamic Measurement, Ministry of Education, North University of China, Taiyuan 030051, China

**Keywords:** asymmetric Y-type microchannel, buffer area, IDT, adjustable tilted-angle, particle control

## Abstract

The precise control of target particles (20 µm) at different inclination angles *θ_i_* is achieved by combining a perturbed asymmetric sheathless Y-type microchannel and a digital transducer (IDT). The offset single-row micropillar array with the buffer area can not only concentrate large and small particles in a fixed region of the flow channel, but also avoid the large deflection of some small particles at the end of the array. The addition of the buffer area can effectively improve the separation purity of the chip. By exploring the manufacturing process of the microchannel substrate, an adjustable tilted-angle scheme is proposed. The use of ataTSAW makes the acoustic field area in the microchannel have no corner effect region. Through experiments, when the signal source frequency was 33.6 MHz, and the flow rate was 20 µL/min, our designed chip could capture 20 µm particles when *θ_i_* = 5°. The deflection of 20 µm particles can be realized when *θ_i_* = 15°–45°. The precise dynamic separation of 20 µm particles can be achieved when *θ_i_* = 25°–45°, and the separation purity and efficiency were 97% and 100%, respectively.

## 1. Introduction

In recent years, the design and application of biochips based on microfluidic technologies have attracted extensive attention from researchers in China and elsewhere [1,2,3]. Among them, the microfluidic separation technology has become an effective tool for cell detections [4,5] and disease diagnosis [6,7]. For example, microfluidic separation chips can be used to separate trace circulating tumor cells in the human peripheral blood for disease diagnosis [8]. The microfluidic separation technology is mainly divided into two categories, namely, passive separation [9,10,11] and active separation [12,13,14]. The former is usually separated by designing different channel geometries to make particles reach different positions at high speeds [15], which has the advantages of high flux, simple design, and low cost. However, a high flow rate may reduce cell activity and even cause cell death. The latter includes electrophoresis [16,17], magnetic separation [18,19], acoustic separation [20,21], and thermal and pneumatic controls [22,23]. Among them, the acoustic control technology has become an ideal choice for manufacturing bioseparation chips due to its advantages of good biocompatibility, low power consumption, simple manufacture, and easy control.

The preprocessing operation is carried out before the acoustic manipulation of cells. Its main purpose is to enable the target cells to be effectively controlled when they enter the acoustic manipulation area. Pretreatment can be divided into two types. Shuaiguo Zhao et al. [24] enabled cell flow at the central line of the channel by increasing sheath flow on both sides of the entrance of the microchannel. The bacteria were isolated from human red blood cells (RBCS) with 96% purity when passing through the sound-controlled area. Sen Xue et al. [25] also achieved the 96% separation of 15 µm target particles in this way. However, the addition of sheath flows increases the design cost and experimental complexity, and may damage cell activity or even cause contamination. In addition, the general sheath flow velocity is higher than the cell solution velocity, so the deflection resistance of the target cells is large, which may reduce the separation efficiency of the chip. Therefore, the use of a sheath-free system is favored by researchers. K. Mutafopulos et al. [26] reported a microfluidic fluorescently activated cell sorting (μFACS) device that performs sorting after focusing cells by inertial flow. They sorted three different cell lines at speeds of several kilohertz, more than 1 meter per second, while maintaining sorting purity and cell viability at around 90%. Only cells larger than 10 μm in diameter could be focused in their experiments, which would limit their clinical application. The most abundant blood cell in human blood is RBCS, which is about 6 μm in diameter [27]. Trace circulating tumor cells (CTCs) in the peripheral blood of patients with early cancer are 14–25 μm in diameter [28]. To isolate CTCs from the peripheral-blood samples of cancer patients, the effective control of other blood cells with smaller diameters is necessary. Therefore, it is of great significance to design a sheath-free chip that can control cells of different diameters.

Sound control technology is mainly divided into two categories, namely, standing surface acoustic waves (SSAWs) [29,30] and traveling surface acoustic waves (TSAW) [31,32]. The former requires at least a pair of IDTs to realize the control. Compared with the latter, the system is more complex, and the operation is complicated. In previous studies, the separation purity of the latter was generally lower than that of the former due to the limitation of the angular effect region. However, Husnain Ahmed et al. in 2018 proposed a sheathless focusing and separation chip for particles in continuous flow based on tilted-angle traveling surface acoustic waves (taTSAWs) [33]. The chip achieved particle focusing and separation by placing two IDTs at ±30°, and achieved high-purity separation of more than 99% at the outlet. This enables high-purity separation based on TSAW. However, their work requires two signal sources to focus and separate particles, which complicates the system. Research on the tilt angle of tilted-angle standing surface acoustic waves (taSSAWs) currently focuses on 5° to 45° [34,35,36]. However, there are few studies on the tilt angle of TSAWs. Therefore, the research of the particle separation chip based on taTSAW is of great significance.

This paper introduces a microfluidic device based on ataTSAW that combines a perturbed sheathless asymmetric Y-type microchannel and an IDT. The offset micropillar array in the microchannel can concentrate large and small particles within a range of 250 µm from the lower wall of the microchannel. We added the buffer area at the end of the offset micropillar array. The buffer area can avoid the large deflection of some small particles at the end of the array and improve the separation purity of the chip. The design of ataTSAWs enables the acoustic field to act on all particles in the microchannel without considering the existence of the angle effect region. When an SAW enter the fluid, a complex fluid–structure coupling effect occurs, forming a leakage SAW wave. The wave enters the fluid at an angle known as the Rayleigh angle (21.8°). The flow channel area corresponding to the angle between SAW and flow channel is less than 21.8° and is not affected by acoustic signals, that is, the angle effect area is formed. A new microchannel substrate preparation process was adopted to realize the adjustable position of the microchannel, so as to test different inclination angles. Compared with other taTSAW systems, the realization of an adjustable inclination angle greatly reduces the experimental cost because there is no need to manufacture multiple chips. Compared with the sheath system, the experimental equipment is relatively simple, has lower cost, and is less likely to cause pollution. Experiments were carried out after the theoretical explanation of the working mechanism of the chip. Experimental results show that our designed chip can effectively control the target particles. When the signal source frequency was 33.6 MHz, and the flow rate was 20 µL/min, the chip could capture target particles when *θ_i_* = 5°. The deflection of target particles could be realized when *θ_i_* = 15°–45°. The precise dynamic separation of target particles could be achieved when *θ_i_* = 25°–45°, and the separation purity and efficiency were 97% and 100%, respectively.

## 2. Materials and Methods

### 2.1. Working Mechanism

An asymmetrical Y-shaped microchannel with polydimethylsiloxane (PDMS) (The Dow Chemical Company, United States) thin film as the sealing layer was attached to 128°Y-X piezoelectric lithium niobate (LiNbO_3_ (Huaying, China) at a certain angle, as shown in Figure 1a. A physical picture of the chip is shown in Figure 1b. The electrodes in the microchannel spanned 73% of the width of the microchannel, which ensured that large particles could move forward to the upper outlet when moving near the upper wall (the closest distance between the electrodes and the upper wall of the flow channel was 135 µm). The realization of different *θ_i_* is through one-to-one correspondence between microchannel locating points and substrate locating points. With the locating points, we aimed the chip with a magnifying glass and tweezers. A schematic diagram of the locating points on the microchannel and substrate are shown in Appendix A. When the IDT inclination angle was *θ_i_*, the acoustic radiation force (*F_ARF_*) (the direction indicated by the arrow in Figure 1c) and stoke resistance (*F_d_*) (the direction indicated by the arrow in Figure 1c) affected the particle movement in the channel. The effect of acoustic waves on polystyrene particles in the acoustic control region can be explained by dimensionless parameter *κ*. When *κ* > 1, the particle is affected by *F_ARF_*, but when *κ* < 1, *F_ARF_* does not dominate the particle motion. The value of *κ* = π*fa*/*c_f_* depends on the frequency of ataTSAW (*f*), particle diameter (*a*), and the speed of sound in the fluid (*c_f_*) [37]. This makes particle separation based on particle size possible.

As shown in Figure 1c, the asymmetrical Y-shaped microchannel consisted of one inlet and two outlets. A mixture containing different particles (blue and red represent large and small particles, respectively) was pumped through the inlet. The particles converged within the range of 250 µm from the lower wall of the channel when they left the micropillar array. Each micropillar was an equilateral triangle with a side length of 30 µm. Yanjuan Wang et al. [38] designed two different shapes of micropillars, circular and triangular. The performance of the two chips was analyzed in terms of stability, flux, recovery efficiency, purity, and enrichment coefficient. Their research showed that triangular chips outperform circular ones. Unlike previous work [39], we added the buffer area (gray triangular micropillar area in Figure 1c) at the end of the micropillar array. The buffer area could increase the velocity of the region above the end of the array, which avoided the large deflection of a few small particles. The relationship between the maximal micropillar gap (Smax) and a is as follows: Smax = 2a [39]. The minimal particle diameter of this test was 5 µm, so the micropillar gap was set to 10 µm.

In Figure 1c, *F*_1_ and *F*_2_ are components of the acoustic radiation force. The direction of *F*_1_ was parallel to *F_d_*, and the direction of *F*_2_ was perpendicular to *F_d_*. The resultant force of *F*_1_ = *F_ARF_*cos *θ_i_* and *F_d_* determined the horizontal force of the particle. Component force *F*_2_ = *F_ARF_*sin *θ_i_* determined the force of the particle in the vertical direction. The resultant force of the horizontal force and vertical force of the particle was its final force *F_R_*. *θ_i_* was 5°–45° in this design. As the angle increased, the value of cos *θ_i_* decreased, and the value of sin *θ_i_* increased. Therefore, as the angle increased, component *F*_1_ decreased, and component *F*_2_ increased. When the angle was small, component force *F*_1_ was large, and *F*_2_ was small. The particles affected by the acoustic radiation force could be captured in the part of the flow channel, where forces *F*_1_ and *F_d_* were equal in magnitude, and force *F*_2_ was almost zero. Force *F_R_* was almost zero; thus, the particle capture could be achieved (as shown in the diagram of the particle force on the bottom left in Figure 1c). With the increase in angle, when *F*_1_ > *F_d_*, and *F*_2_ was a certain value, force *F_R_* is shown in the diagram of the particle force on the bottom middle in Figure 1c. At that point, the particle could be deflected. As the angle continued to increase, as shown in the force diagram of the particle on the bottom right in Figure 1c, force *F_R_* increased significantly. In this case, the particle had to be able to achieve deflection, and the effect was better than that of the former.

When large particles moved into an effective acoustic field area, the *κ* of large particles was greater than 1, so large particles moved along the direction of the resultant force (*F_R_*) in Figure 1c under the combined action of *F_ARF_* and *F_d_*. The motion of the large particles depended on the magnitude and direction of the *F_R_*. When the angle was set to deflect large particles away from the effective sound field area, *F_d_* directed the particles out of the upper outlet (blue particles in Figure 1c). When small particles moved into the effective acoustic field area, the *κ* of small particles was smaller than 1, so *F_d_* mainly led the particle movement until it flowed out of the lower outlet (as shown by red particles in Figure 1c). By controlling the frequency of ataTSAW, *κ* > 1 for large particles and *κ* < 1 for small particles were realized, which enabled the deflection separation of large particles, while the small particles were not affected when the particles passed through the acoustic field area. *θ_i_* affected the motion state of large particles. If *θ_i_* was too small, particle capture could be achieved, but particle deflection was difficult to achieve. With the increase in *θ_i_*, the particle deflection effect was different. Therefore, it is meaningful to explore the motion states of particles under different *θ_i_*. Previous research on *θ_i_* mainly focused on 30° [33]. We tested the control effect of the electrodes on large particles when *θ_i_* was from 5° to 45°.

Micromachined IDTs were used as wave generators in this study. Studies showed that 20 μm polystyrene particles can achieve optimal deflection when *κ* = 1.47 [39]. Applying radio frequency (RF) signals of 33.6 MHz to IDTs produces TSAWs to control the large particles (ee Appendix A for the characteristic frequency simulation diagram of SAW devices, and Appendix A for the substrate surface wave velocity at different frequencies). The parameters of IDTs in this work were designed as follows: acoustic aperture was 800 μm, and finger spacing and finger width were 29 μm. An IDT consisting of a pair of cross-finger electrodes was used in this study. In this setting, *κ* for 20 μm particles was 1.47, and *κ* for 5 μm particles was 0.37.

### 2.2. Chip Fabrication 

Our chip manufacturing process flowchart is shown in Figure 2. We used standard soft-lithography techniques with SU8 master mode on silicon substrate. First, a positive photoresist (AZ6130, compound semiconductor material (Wuxi Huarun Crystal Chip Semiconductor Co., Ltd., Wuxi, China) was rotated onto a 4-inch silicon wafer. Ultraviolet (UV) exposure was performed after baking. After exposure, the developing operation was carried out, and after microscopic examination, the film was baked on a drying table at 120 °C. Then, a 40 µm thick flow channel structure was prepared on the silicon wafer with a deep silicon etching process to complete the preparation of the silicon mold. According to the PDMS:curing agent = 10:1 configuration ratio; the two were evenly mixed and poured into the silicon mold. After baking at 75 °C for 45 min, it was peeled off and cut into corresponding shapes. A thin layer of PDMS mixture was rotated onto the single-side polished glass and baked until it had been shaped. Through experiments, we found that, when the speed was set at the low level of 500 r/s and the high level of 2000 r/s, it was not only difficult for the PDMS film to leak liquid in the flow channel, but it also obstructed little the acoustic signal. At that point, the thickness (*t*) of the PDMS thin layer was 50 µm. The wavelength (λ) of the device was 116 µm. When *t* < 2λ, the ARF effect dominated the acoustothermal effect inside the microchannel [40]. The bonded surface of the PDMS microchannel and glass was treated with a plasma process. The two were quickly bonded to achieve irreversible bonding, and stripping and cutting were then completed to fabricate the position-adjustable microchannel. The TSAW electrode substrate separator was prepared with an ion beam etching (IBE) process. First, Cr/Au (20 + 100 nm) was sputtered onto LiNbO_3_. Then spin positive photoresist (AZ6130) on its surface. After UV exposure and development, the film was heated and hardened. The IDT electrodes were prepared by etching the substrate with IBE for 5 min. The etched model was then placed in acetone solution for ultrasonic degumming. Lastly, the fabricated microchannel was reversibly bonded with LiNbO_3_ to produce a complete device.

### 2.3. Sample Preparation and System Setup

In this experiment, two kinds of particles were selected for experimental testing. Red blood cells are about 6 µm in diameter [27]. Therefore, 5 µm polystyrene fluorescent microspheres were used to simulate the blood cells. Circulating tumor cells in the peripheral blood of patients with early cancer are about 14–25 µm [28], so 20 µm polystyrene microspheres were used to simulate circulating tumor cells. Reagent 1 was configured with 10 µL, 5 µm monodisperse polystyrene fluorescent microsphere particle suspension (5 µm, 10 mL, 1 wt%, Wuxi Regal Biotechnology Co., Ltd., Wuxi, China), 0.1 mL Tween-20 (Sevenbio, Beijing, China) and 4 mL deionized water. Reagent 2, containing 20 μm monodisperse polystyrene fluorescent microsphere particle suspension (20 µm, 10 mL, 1 wt %, Wuxi Regal Biotechnology Co., Ltd.,), was configured in the same way. Under fluorescence excitation (the excitation wavelength was 400–410 nm), 5 and 20 µm particles appeared to be green and blue, respectively. The circular tube containing the prepared solution was placed inside an ultrasonic oscillator (BKE-1002DT, Hangzhou Boke Ultrasonic Equipment Co., Ltd., Hangzhou, China) to oscillate for 10 min to evenly distribute the suspended particles in the solution. The solution was subsequently loaded into the microsyringe (Luer interface injection, 5 mL, Shanghai Conde Lai Enterprise Development Group Co., Ltd., Shanghai, China) and pumped into the microfluidic chip through a Tygon tube (CN-06419-03, Shanghai Conde Lai Enterprise Development Group Co., Ltd.,). The microsyringe was connected to the Tygon tube through a flat-mouthed needle (0.8 mm outside diameter; Shanghai Conde Lai Enterprise Development Group Co., Ltd.,). The Tygon tube was connected to the chip inlet through a steel needle (0.8 mm outside diameter; Shanghai Conde Lai Enterprise Development Group Co., Ltd.,). The fluid flow was powered by setting the flow rate of the microinjection pump (one-channel microsyringe pump, XMSP-1C, Ximai, Nanjing, China). A function signal generator (FY6900-60M, FeelElec, Zhengzhou, China) was used to output sine wave signals of fixed frequency and size. The particles were tracked with a fluorescence microscope (NIB620-FL, Nexcope, Ningbo, China) and a 6.3 million pixels high-speed CMOS camera (Nexcam-TC6CCD, Nexcope, Ningbo, China) to record the movements of the particles. A blood cell counter (automatic cell counter, C100,Rayward Life Technology Co., Ltd., Shenzhen, China) was used to measure the type and number of particles in the solution collected at different outlets. At least three sampling tests were required for each outlet.

## 3. Results

### 3.1. Simulation of Two-Dimensional (2D) Surface Velocity and Three-Dimensional (3D) Cross-Sectional Velocity of Microchannels

The sample flow was injected into the microfluidic chip at a certain speed. As the particles flowed through the micropillar array, they moved along the offset direction of the micropillar array. The velocity distribution of the flow field at the end of the micropillar array affects the particle trajectory. Therefore, it is of great significance to study the arrangement of micropillar arrays. COMSOL software was used to simulate the 2D surface velocity and 3D cross-sectional velocity of microchannels. In order to verify the effect of the optimized micropillar array on the velocity distribution of the flow field, the velocity distribution in the micropillar array with and without a buffer area was compared. Considering the impact of flow rate on acoustic control, we set the overall working flow rate at 20 µL/min. Our previous work demonstrated the effectiveness of acoustic manipulation of particles at this velocity [39]. 

The results show that there was no significant speed increase above the end of the micropillar array without the buffer area (Figure 3a). Above the end of the micropillar array with the buffer area, the effect of the speed increase at the end of the micropillar was achieved due to the narrowing of the upper region (Figure 3b). Figure 3c shows the flow field distribution of the cross-section of the microchannel with the buffer area at partial positions of the micropillar array. When the fluid moved to Position 1 in Figure 3c (the same position at the end of the micropillar in Figure 3a), the flow rate below the micropillar array (the left area of the micropillar array in Figure 3c) was significantly higher than the flow rate above the micropillar array (the right area of the micropillar array in Figure 3c). If there is no buffer area, some small particles may have large deflection and discharge from the upper outlet to reduce the separation purity of the chip. The velocity distribution at the end of the buffer area is shown in Position 2 in Figure 3c (the same position as the end of the micropillar in Figure 3b). There was an obvious speed increase above the micropillar array, which avoided the large deflections of some small particles below the micropillar array, and thus improved the separation purity of the chip. The optimized micropillar array was tested and validated as shown below.

### 3.2. Related Performance Testing of Microchannels When AtaTSAWs Off

First, the particle motion at the end of the micropillar and the outlet of the flow channel without signal was measured. The chip was placed above the electrodes at any angle for testing (*θ_i_* is 25° in Figure 4). Obviously, because the signal was off, the particle trajectory was independent of *θ_i_* in this case. The effect picture of particle motion track superimposed is shown in Figure 4. Both 5 and 20 μm particles converged within 250 μm from the lower wall of the flow channel (Figure 4a, Appendix A). Large (20 μm) and small (5 μm) particles had no large deflection at the end of the micropillar array due to the effect of the buffer area. The outlet width of the flow channel was set as the upper outlet (150 μm) and lower outlet (350 μm) = 3:7. When ataTASWs were off, both 5 and 20 μm particles flowed out of the lower outlet (Figure 4b, Appendix A). 

The blood cell counter was used to test three samples from different outlets. The solution volume of each sample was 0.1 mL. The data of sampling test results are shown in Table 1. Among them, the test results of the first sampling from the upper and lower outlets are shown in Figure 4c,d, respectively. There were no particles in the upper outlet (Figure 4c), and there were 5 and 20 μm particles in the lower outlet (Figure 4d). The optimized design of micropillar is, therefore, reasonable, and the design of the outlet width of the flow channel could realize all particles being able to flow out of the lower outlet when ataTSAWs were off.

### 3.3. Acoustic Separation Effect Test at Different θ_i_

The movable flow channel could realize the combination of different *θ_i_* between the flow channel and electrodes. We tested the motion state of particles with *θ_i_* of 5°, 15°, 25°, 35°, and 45° at the signal source frequency of 33.6 MHz. The test pictures after the superposition of the motion track of the particles are shown in Figure 5. When *θ_i_* = 5°, the capture of 20 μm particles could be achieved, and 5 μm particles kept their original trajectory (Figure 5a, Appendix A). This is because, when the *θ_i_* was too small, the component *F*_1_ (Figure 1c) of *F_ARF_* was enough to be equivalent to *F_d_*, and the component *F*_2_ (Figure 1c) of *F_ARF_* was almost zero, so the capture of 20 μm particles could be achieved in a certain position, and almost no deflection occurred. At this angle, only 5 μm particles flowed out of the lower outlet, and no particles flowed out of the upper outlet (Figure 5f, Appendix A). Figure 5b–e show that, when *θ_i_* = 15°–45°, all 20 μm particles could realize deflection and move forward. When *θ_i_* = 15°, the maximal deflection of 20 μm particles was about 130 μm away from the upper wall of the flow channel (Figure 5b, Appendix A). At this angle, component *F*_2_ was not large enough to push 20 μm particles to reach the upper wall, so 20 μm particles flowed out from the lower outlet (Figure 5g). When *θ_i_* = 25°–45°, 20 μm particles could reach the upper wall under the action of *F_ARF_* (Figure 5c–e, Appendix A). With the increase in *θ_i_*, the position of the acoustic field area (gray bar area in Figure 5a–e) changed, leading to a gradual backward shift of the 20 μm particles’ deflection position. When *θ_i_* was 25°, 35°, and 45°, the maximal lateral migration distances required for 20 μm particles to complete the deflection were 752, 907, and 1020 μm, respectively (Figure 5c–e). With the increase in *θ_i_*, the maximal lateral offset distance required for the 20 μm particles to complete the deflection was enlarged due to the decrease in the effective acoustic field area (the red dotted box area in Figure 5a–e). When *θ_i_* = 25°–45°, 5 μm particles flowed out of the lower outlet, and 20 μm particles flowed out of the upper outlet (Figure 5h, Appendix A).

In conclusion, the chip that we designed could control the target particles (20 µm) from different angles. It could capture 20 µm particles when *θ_i_* = 5°. Gina Greco et al. [41] presented a simple surface acoustic wave (SAW)-based platform for dynamic cell culture that was compatible with standard light microscopes, incubators, and cell culture dishes. Compared with the standard static culture, the cell proliferation rate increased by 36 + 12% under this condition. The presence of SAWs did not increase cell death, and cell morphology did not change. In our experiments, *θ_i_* = 5° could be used to improve cell proliferation, which supports the great versatility and biocompatibility of the SAW technology for cell manipulation. The deflection of 20 µm particles could be realized when *θ_i_* = 15°–45°. The precise dynamic separation of 20 µm particles could be achieved when *θ_i_* = 25°–45°. The precise dynamic separation of particles indicates that our chip could be used for cell sorting. For example, it could be used to collect small amounts of circulating tumor cells from the peripheral blood of patients with early-stage cancer to facilitate detection [27], diagnosis, and treatment.

### 3.4. Test Results of Chip Separation Performance When θ_i_ = 25°–45°

When *θ_i_* = 25°–45°, and the signal source frequency was 33.6 MHz, 20 μm particles flowed out of the upper outlet, and 5 μm particles flowed out of the lower outlet. The separation purity of the chip was characterized by testing the content of 5 μm (nontarget) particles in the upper outlet. The separation efficiency of the chip was characterized by testing the content of 20 μm (target) particles in the lower outlet. At different angles, the upper and lower outlets were sampled for three times and then tested with a blood cell counter. The solution volume of each sample was 0.1 mL. The test results of chip separation purity and efficiency are shown in Figure 6a,b, respectively. When *θ_i_* = 25°–45°, the separation purity of the chip remained above 97% (the scatter diagram is shown in Figure 6a). The content selected in the black, red, and blue dotted boxes is the particle type and content of a sample at the upper outlet when *θ_i_* was 25°, 35°, and 45°, respectively (Figure 6a). When *θ_i_* = 25°–45°, the separation efficiency of the chip remained at 100% (the scatter diagram is shown in Figure 6b). The content selected in the black, red, and blue doted boxes is the particle type and content of a sample at the lower outlet when *θ_i_* was 25°, 35°, and 45°, respectively (Figure 6b). In conclusion, when *θ_i_* = 25°–45°, the separation purity and efficiency of the chip were up to 97% and 100%, respectively.

## 4. Discussion

We compare this chip with a passive separation chip [15], and chips based on TSAW [28], SSAW [30], taTSAW [33], and taSSAW [24]. The comparison is shown in Table 2.

Table 2 shows that the passive separation chip [15] requires higher flow rates. Higher flow rates can avoid the use of sheath flow, but may also greatly reduce cell activity. Therefore, the passive separation chip has limitations in the application of biological cell separation. Its separation purity is lower than that of the active sorting chip, which cannot meet the needs of high-precision separation. Compared with the TSAW-based separation chip [28], our work significantly improves the separation purity and does not require sheath flow. Compared with the SSAW-based sorting chip [30], our chip reduces the use of IDTs and does not require sheath flow. Compared with the chip based on taSSAW [24], our chip requires only one IDT and no sheath flow. The use of sheath flow can increase the experimental complexity due to the need for multiple pumps, and may lead to sample contamination. Compared with the currently developed taTSAW chip [33], our chip reduces the use of IDTs due to the addition of micropillar array, and carries out a more specific multangular study. In addition, we used a microchannel chip and an IDT to complete all angle tests during the whole experiment. This indicates that the repeated rebonding of the microchannel with the IDT did not cause mechanical damage to the IDT. This shows the robustness of our chip and the high utilization rate of the IDT.

In an ordinary laboratory environment, the adhesion strength of the microchannel and the substrate is reduced by repeated bonding. When the adhesion strength decreases to a certain value, the ARF would subsequently be too weak to control cells. In order to obtain the effective bonding times of the chip, we conducted five repeated sorting experiments when *θ_i_* was 35°. After 20, 40, 60, 80, and 100 repetitions of the chip, the maximal lateral migration distance required by the particle to complete the deflection was tested. The maximal lateral migration distance required to complete the deflection of the test particle was used to characterize the acoustic attenuation caused by repeated bonding. The test results are shown in Table 3. Our chip could still produce the same result within 80 times of repeated bonding with an error of less than 1% (compared with Figure 5d).

## 5. Conclusions

This paper presented a sheathless microfluidic device based on ataTSAWs. The device achieved the precise control of target particles by combining a perturbed asymmetric Y-shaped microchannel with an IDT at a certain angle. The perturbation structure was an array of offset micropillars with the buffer area. The addition of the buffer area could avoid the large deflection of some small particles (5 μm) at the end of array, thus improving the purity of the chip. A microchannel with PDMS film as the sealing layer was adopted to render the flow channel mobile. By setting locating points on the microchannel and the LiNbO_3_ substrate, the control of the electrodes on the target particles at different inclination angles was studied. This design greatly improved the recovery efficiency of the electrode. When the signal source frequency was 33.6 MHz, the designed electrode could control 20 μm particles in a mixture of 5 and 20 μm particles. When *θ_i_* = 5°, the capture of 20 μm particles could be achieved. When *θ_i_* = 15°–45°, the deflection of 20 μm particles could be achieved. When *θ_i_* = 25°–45°, the precise dynamic separation of 20 μm particles could be achieved, and separation purity and efficiency were 97% and 100%, respectively. This work shows an improved scheme for improving the separation purity of chips. The preparation technology of adjustable-angle microchannels provides a new idea for multangular research. This conclusion can provide reference for free particle manipulation, and is expected to be used in biological cell detection and analysis.

## Figures and Tables

**Figure 1 biosensors-12-00611-f001:**
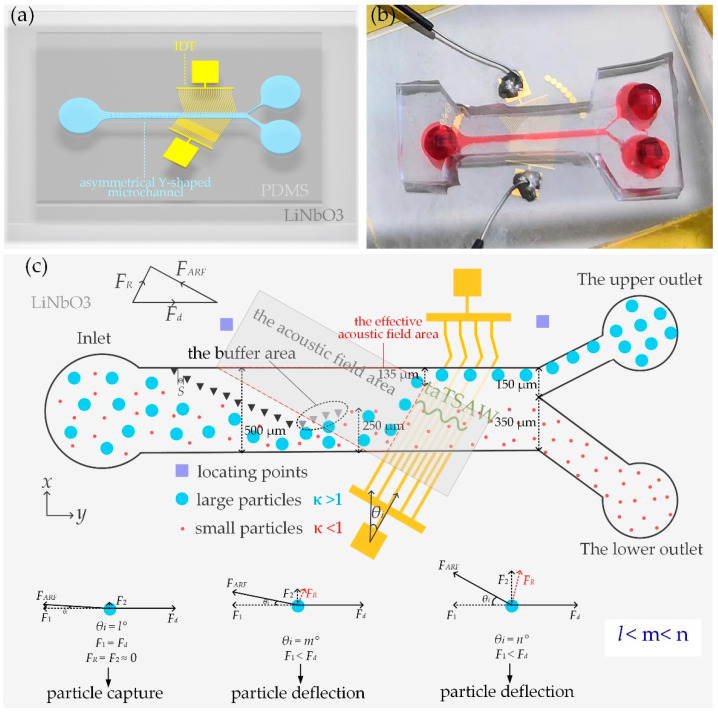
(**a**) Overall schematic diagram of the chip; (**b**) physical picture of the chip; (**c**) schematic diagram of sheathless particle separation between coupled micropillar array and ataTSAWs. The micropillar array concentrates all the particles in a region 250 µm away from the lower wall of the flow channel. The IDT at a certain angle of *θ_i_* enabled the deflection separation of large particles, while small particles were not affected when the particles passed through the acoustic control region.

**Figure 2 biosensors-12-00611-f002:**
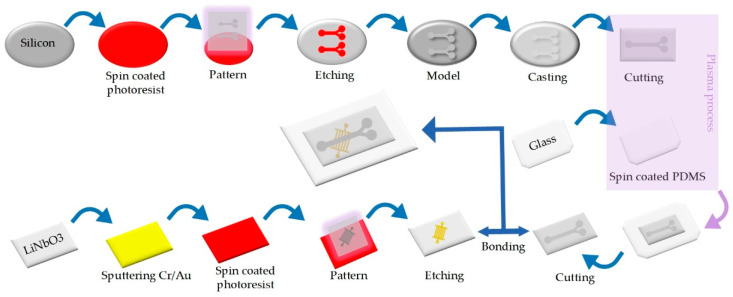
Flowchart of chip manufacturing process.

**Figure 3 biosensors-12-00611-f003:**
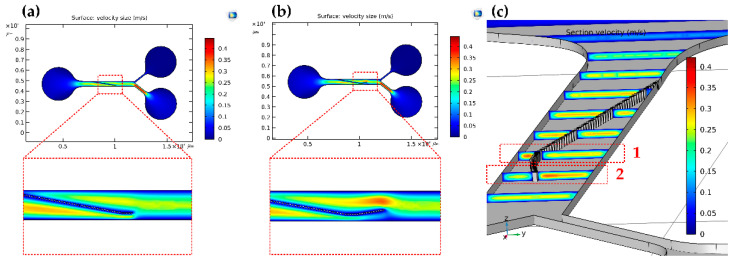
(**a**) Velocity distribution in the micropillar array without a buffer area; (**b**) velocity distribution in the micropillar array with a buffer area; (**c**) flow field distribution of the cross-section of the microchannel with the buffer area.

**Figure 4 biosensors-12-00611-f004:**
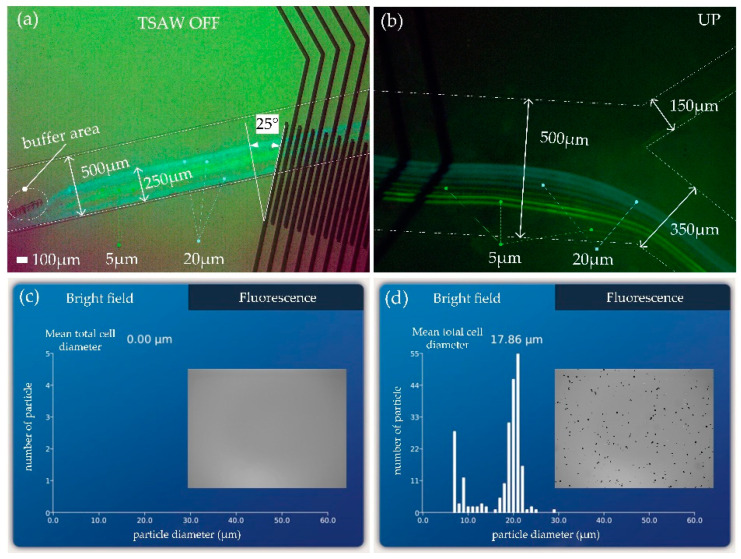
(**a**) Particle trajectories at the end of the buffer area; (**b**) particle trajectories at the outlet of the flow channel; (**c**) test results of various particle types and numbers in the upper outlet; (**d**) test results of various particle types and numbers in the lower outlet.

**Figure 5 biosensors-12-00611-f005:**
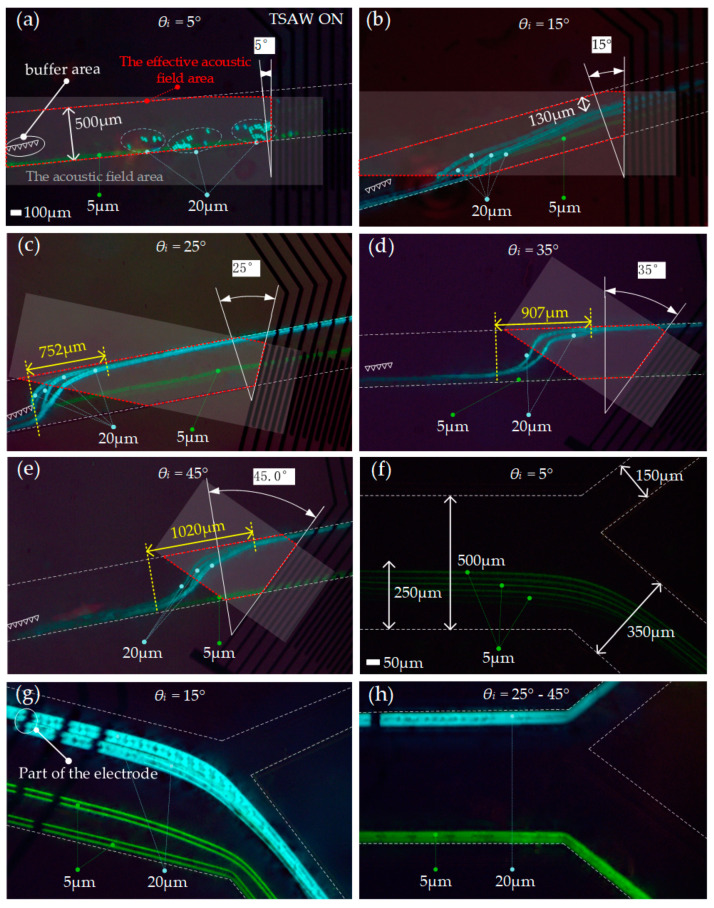
Test diagram of particle track at different positions under different *θ_i_*: Track test diagram of particles in the action region of a sound field when (**a)** *θ_i_* = 5°, (**b**) *θ_i_* = 15°, (**c**) *θ_i_* = 25°, (**d**) *θ_i_* = 35°, and (**e**) *θ_i_* = 45°; track test diagram of particles at the outlet of the flow channel when (**f**) *θ_i_* = 5°, (**g**) *θ_i_* = 15°, and (**h**) *θ_i_* = 25°–45°.

**Figure 6 biosensors-12-00611-f006:**
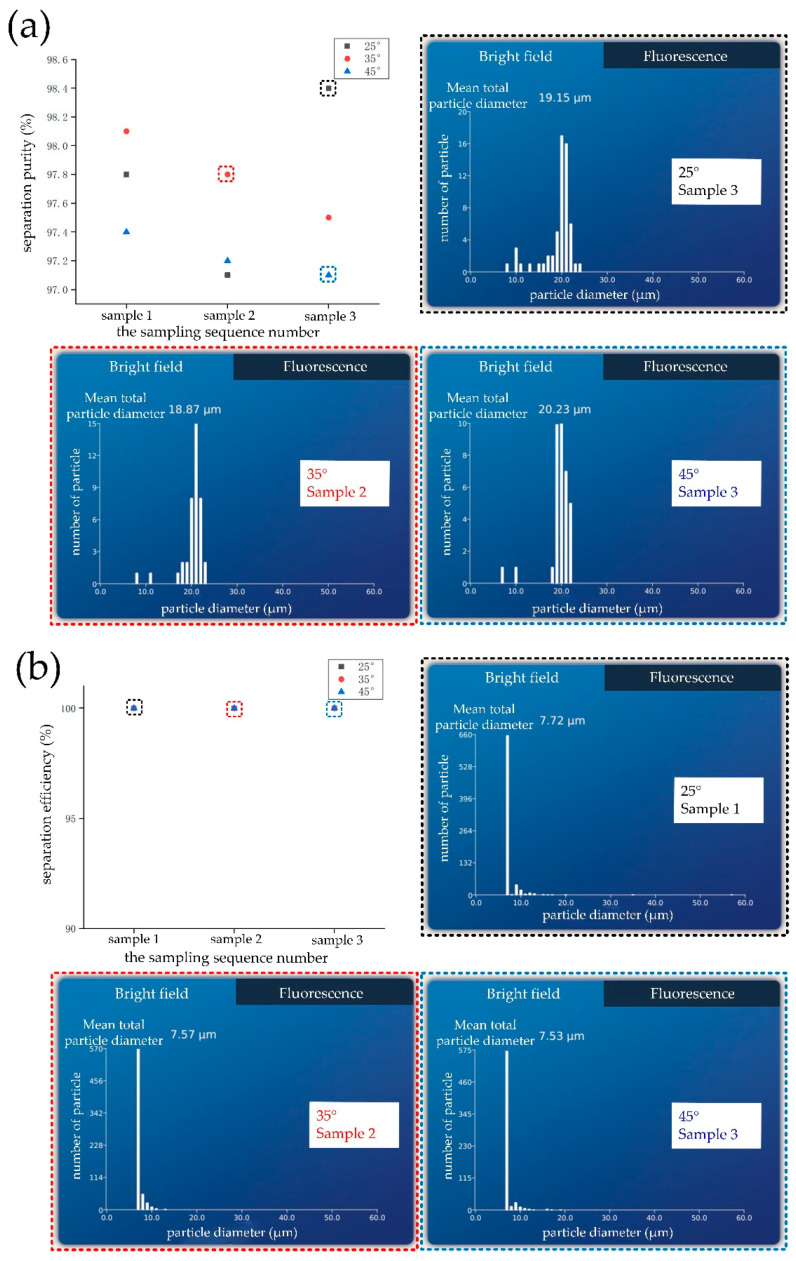
Test results of chip separation performance: (**a**) Test result diagram of chip separation purity; (**b**) test result diagram of chip separation efficiency.

**Table 1 biosensors-12-00611-t001:** Test statistics of particles at different outlets when ataTSAWs were off.

Outlet Type	Sample Number	5 μm Particle Number	20 μm Particle Number	Total Number of Particles
Upper outlet	Sample 1	0	0	0
	Sample 2	0	0	0
	Sample 3	0	0	0
Lower outlet	Sample 1	123	42	165
	Sample 2	101	27	128
	Sample 3	115	30	145

**Table 2 biosensors-12-00611-t002:** Comparison list of different chips.

Chip Type	Flow Velocity	Separation Purity	Tilt Angle	IDT Number	Require Sheath Flow	Can Electrodes Be Reused?
[15]	4 mL/min	93.59%	No	No	No	No
[28]	25 µL/min	90%	No	1	Yes	No
[30]	67.5 µL/min	92.7%	No	2	Yes	No
[33]	50 µL/min	99%	±30°	2	No	No
[24]	6 µL/min	96%	15°	2	Yes	No
Our work	20 µL/min	97%	5–45°	1	No	Yes

**Table 3 biosensors-12-00611-t003:** Results of repeated tests.

Number of Repeated Bonding	Maximal Lateral Migration Distance (µm)	Error Range (Compared with Figure 5d)
20	908	0.1%
40	911	0.4%
60	913	0.7%
80	916	1%
100	920	1.4%

## Data Availability

Not applicable.

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
