# Peer review of "A Perturbed Asymmetrical Y-TypeSheathless Chip for Particle Control Based on Adjustable Tilted-Angle Traveling Surface Acoustic Waves (ataTSAWs)"

_biosensors, 2022, doi:10.3390/bios12080611_

Round 1
Reviewer 1 Report
In this work, a precise control of the target particles is achieved by combining a perturbed asymmetric sheathless Y-type microchannel and the ataTSAW. Micropillars with the buffer area was designed to affect the particle trajectory and thus improve the purity. The whole manuscript is well structured and easy to read. However, the following comment should be considered before publication.
1. The comparison work between standing surface acoustic waves (SSAW) and traveling surface acoustic waves (TSAW) should be given in depth. And the parameter design of IDTs in this work should be given.
2. Figure 1b shows obvious leakage around the inlet and both outlets. Will it affect the sorting process? How is the bonding quality of PDMS chip with glass substrate by plasma treatment? And how is the liquid sample injected into the channel with a certain flow rate since no pump or connector was shown in figure 1b. Such description should be given.
3. How is the fabrication quality of micropillar arrays in microchannel, such as an SEM image in the morphology of micropillar. Besides, In SAW sorting application, the SAW would definitely affect the flow field distribution or velocity distribution .
4. The mean total cell diameter in figure 4c and 4d showed that the diameter of the smaller particle was roughly 10 μm, rather than 5 μm that was mentioned in the main text.
5. How long does it require for sample 1 / 2 / 3 to sort 20 μm particle from the mixture sample?
6. The resolution of Figure 6a, 6b should be improved since the review cannot see the details.
7. The review is confused about the definition of IDT number. A pair of IDTs is shown in Figure 1, which seems inconsistent with what is shown in Table 2.
Reviewer 2 Report
In the manuscript, the authors report a surface acoustic wave microfluidic device for particle separation. The authors employed the Tilted-angle Traveling Surface Acoustic Waves (ataTSAW) and perturbed asymmetric sheathless Y-type microchannel for improving the separation. By tuning the angle of ataTSAW, they could optimize the device to achieve the separation purity of 97% and efficiency of 100%, separately. I think this is a fine design and application of the acoustofluidic separation. The author demonstrated the device works, and the results make sense. However, the data is very preliminary and the writing needs to be significantly improved. Thus, I would recommend accepting this work after addressing some major aspects below:
1) The authors didn't discuss the particle separation based on bulk acoustic waves, which is the other very important acoustic separation along with the surface acoustic wave-based method.
2) Through reading the introduction, I think the innovation of this work is unclear once compared with recent similar work (Anal. Chem. 2018, 90, 14, 8546–8552).
3) The authors showed the simulation of the velocity profile within the microfluidic channel with micropillar arrays. However, there are no experimental data to match. The authors need to show both simulation and experimental results of the trajectory of the particle under different flow rates and determine the best condition for acoustic separation.
4) Figure 6 is not readable. The authors need to improve this figure.
5) The tests on device-to-device variation are missing. It is hard to evaluate the robustness of this approach.
7) The recovery rate also needs to be discussed, tested, and compared with other acoustic separation methods.
Reviewer 3 Report
Please find attached reviewer comments.

Round 2
Reviewer 1 Report
In line 252-253, “The solution content of each sample was 0.1mL.” In here, the subject should be “solution volume”, rather than “solution content”. A space should be added between the value (0.1) and unit (ml), which actually is 0.1 mL.
The author should list the specific information for the working materials (PDMS, LN substrate) and related facilities such as blood cell counter, optical microscope and a high-speed CCD camera.
Reviewer 3 Report
Please find attached Reviewer Comments.
